# Aitchison Embeddings for Learning Compositional Graph Representations

**Nikolaos Nakis** [1]  **Chrysoula Kosma** [2]  **Panagiotis Promponas** [3]  **Michail Chatzianastasis** [4]  **Giannis Nikolentzos** [5]

## Abstract

Representation learning is central to graph machine learning, powering tasks such as link prediction and node classification. However, most graph embeddings are hard to interpret, offering limited insight into how learned features relate to graph structure. Many networks naturally admit a role-mixture view, where nodes are best described as mixtures over latent archetypal factors. Motivated by this structure, we propose a compositional graph embedding framework grounded in Aitchison geometry, the canonical geometry for comparing mixtures. Nodes are represented as simplex-valued compositions and embedded via isometric log-ratio (ILR) coordinates, which preserve Aitchison distances while enabling unconstrained optimization in Euclidean space. This yields intrinsically interpretable embeddings whose geometry reflects relative trade-offs among archetypes and supports coherent behavior under component restriction; we consider both fixed and learnable ILR bases. Across node classification and link prediction, our method achieves competitive performance with strong baselines while providing explainability by construction rather than post-hoc. Finally, subcompositional coherence enables principled component restriction: removing and renormalizing subsets preserves a well-defined geometry, which we exploit via subcompositional dimensionality removal to probe how archetype groups influence representations and predictions. *Code is available here:* https://github.com/Nicknakis/AICoG

---

[1]Human Nature Lab, Yale University, New Haven, USA [2]Université Paris Saclay, Université Paris Cité, ENS Paris Saclay, CNRS, SSA, INSERM, Centre Borelli, Gif-sur-Yvette, France [3]Yale Institute for Network Science, Yale University, New Haven, USA [4]École Polytechnique, Institut Polytechnique de Paris, Palaiseau, France [5]Department of Informatics and Telecommunications, University of Peloponnese, Peloponnese, Greece. Correspondence to: Nikolaos Nakis <nicolaos.nakis@gmail.com>.

*Proceedings of the $43^{rd}$ International Conference on Machine Learning*, Seoul, South Korea. PMLR 306, 2026. Copyright 2026 by the author(s).

## 1. Introduction

Graph-structured data is ubiquitous across a wide range of application domains, including social networks, telecommunication networks and chemo-informatics. To analyze graph-structured data and uncover meaningful patterns, it is often necessary to apply machine learning methods (Chami et al., 2022). Central to this process is learning high-quality representations of different components of graphs, such as their nodes (Zhou et al., 2022).

A large body of prior work has focused on learning node embeddings that preserve structural proximity in graphs. Early approaches such as DeepWalk (Perozzi et al., 2014) and node2vec (Grover & Leskovec, 2016) adapt techniques from word embedding to random walks, while more recent methods incorporate architectural advances such as attention mechanisms (Abu-El-Haija et al., 2018), multi-context representations (Epasto & Perozzi, 2019), task-specific objectives (Sun et al., 2019; Duong et al., 2023), or non-Euclidean latent spaces (Nickel & Kiela, 2017; 2018). While these methods achieve strong predictive performance, their latent representations typically lack intrinsic interpretability: similarity is encoded geometrically, but distances and directions are not endowed with semantics that support explanation beyond post-hoc analysis, particularly when the latent structure is continuous or overlapping.

Beyond homophily-based similarity, many networks are more naturally characterized by the roles that nodes play. From this perspective, nodes are similar if they exhibit comparable structural or functional behavior, even when they are not adjacent inside the graph. Classical role-based formulations originate from structural equivalence (White et al., 1976) and stochastic block models (SBMs) (Holland et al., 1983), where roles correspond to equivalence classes defined by interaction patterns. Mixed-membership extensions allow nodes to participate in multiple roles, but still assume discrete, identifiable, and axis-aligned latent components. As a consequence, existing role-based embedding methods such as RolX (Henderson et al., 2012), struc2vec (Ribeiro et al., 2017), and GraphWave (Donnat et al., 2018) provide limited intrinsic interpretability when the role structure varies continuously, overlaps across nodes, or is not well approximated by discrete blocks.

In many real-world networks, roles are inherently relative

and context-dependent: nodes are distinguished not by absolute membership in a single role, but by how they trade off multiple latent factors. Standard Euclidean embeddings can represent such a variation, but their geometry does not impose semantics on relative differences between dimensions, making interpretation sensitive to arbitrary coordinate choices. Mixed-membership models allow overlap, but remain tied to axis-aligned latent roles, which is restrictive when role structure is continuous or non-identifiable. These limitations motivate graph embedding frameworks whose latent geometry is explicitly aligned with compositional structure, where similarity depends on relative proportions rather than absolute magnitudes.

Building on this observation, we propose Aitchison Compositional Graph embeddings (AICoG), a role-based graph embedding framework that represents nodes as compositions over latent archetypal factors and compares them using Aitchison geometry (Aitchison, 1982), the canonical geometry for compositional data. Node compositions are embedded via isometric log-ratio (ILR) coordinates, which preserve Aitchison distances while enabling unconstrained optimization in Euclidean space. In contrast to axis-aligned role assignments, AICoG explains roles as regions of a compositional latent space, characterized by stable relative trade-offs among archetypes rather than by individual latent dimensions.

Our contributions are summarized as follows:

**(i)** We introduce Aitchison Compositional Graph embeddings (AICoG), a graph embedding framework that models nodes as compositions and compares them using Aitchison geometry.

**(ii)** We show that embedding compositions via isometric log-ratio (ILR) coordinates yield a latent distance model that is expressively equivalent to Euclidean latent distance models, while endowing distances with principled, invariant semantics grounded in relative trade-offs.

**(iii)** We propose a geometric notion of roles, where roles correspond to regions of a compositional latent space rather than axis-aligned latent dimensions, enabling intrinsic interpretability under continuous and overlapping role structure.

**(iv)** We demonstrate that subcompositional coherence enables principled component restriction: removing and renormalizing subsets of archetypal factors preserves a well-defined geometry, which we exploit to analyze component influence and stability.

**(v)** Through extensive experiments on link prediction and node classification, we show that AICoG achieves competitive predictive performance with strong baselines while providing explainability by construction rather than post-hoc attribution.

## 2. Related Work

**Graph representations in Euclidean spaces.** Many node embedding methods learn representations in unconstrained Euclidean space to preserve structural proximity. These include spectral approaches such as Laplacian Eigenmaps (Belkin & Niyogi, 2001), random-walk-based methods such as DeepWalk (Perozzi et al., 2014) and node2vec (Grover & Leskovec, 2016), and related proximity-preserving techniques such as LINE (Tang et al., 2015). More recently, graph neural networks (GNNs) learn Euclidean embeddings through message passing and neighborhood aggregation (Kipf, 2016; Hamilton et al., 2017; Velickovic et al., 2017). Although effective for prediction, these approaches provide limited intrinsic interpretability of the learned latent geometry.

**Role-based and block-structured graph models.** Roles in networks have traditionally been studied through the notions of structural and stochastic equivalence. Classical models such as the stochastic block model (SBM) (Holland et al., 1983) and its mixed-membership extensions (Airoldi et al., 2007; Jin et al., 2022) represent nodes as distributions over discrete roles or communities, with interactions governed by role–role connectivity patterns. These models provide interpretable role assignments, but rely on identifiable, axis-aligned latent roles, and are best suited to block-structured networks. Structural role embedding methods, including RolX (Henderson et al., 2012), struc2vec (Ribeiro et al., 2017), Role2Vec (Ahmed et al., 2018), and GraphWave (Donnat et al., 2018), aim to capture regular equivalence by embedding nodes with similar structural signatures nearby. While effective for capturing regular equivalence, much prior work implicitly treats roles as discrete types or identifiable axes, i.e., as pure archetypes that nodes approximate or mix over. Our approach instead models roles as continuous, overlapping mixtures and explains similarity through the geometry of relative trade-offs, rather than through axis-aligned role assignments.

**Explainability in graph representation learning.** Explainability in graph learning is often addressed via post-hoc attribution methods (Ying et al., 2019; Yuan et al., 2020). Several works have targeted intrinsic interpretability via architectural bias, including attention mechanisms (Pope et al., 2019), disentangled representations (Baldassarre & Azizpour, 2019), and prototype- or motif-based models (Lin et al., 2020; Zhang et al., 2022; Ying et al., 2021). Related ideas appear in signed and generative graph models (Nakis et al., 2025b;a). These approaches primarily explain predictions or latent factors in Euclidean space, whereas our work endows the latent geometry itself with semantics tailored to compositional structure.

# 3. Proposed Method

**Preliminaries.** Let $\mathcal{G} = (V, E)$ be a simple undirected graph with $N = |V|$ and adjacency matrix $\boldsymbol{Y} \in \{0, 1\}^{N \times N}$, where $\boldsymbol{Y}_{ij} = \boldsymbol{Y}_{ji}$ and $\boldsymbol{Y}_{ii} = 0$. Bold uppercase letters such as $\boldsymbol{X}$ denote matrices, bold lowercase letters such as $\boldsymbol{x}$ denote vectors, and non-bold letters such as $x$ denote scalars. Throughout this work, we adopt a geometric view of interpretability, where semantic meaning is attributed to distances, directions, and regions of the latent space, rather than to individual coordinate axes. We next propose Aitchison Compositional Graph embeddings (AICoG), which models each node as a composition over $K$ latent archetypal components and induces an interpretable latent geometry in Aitchison space (see Figure 1 for an overview).

**Compositional data and geometry.** Compositional data describe relative allocations of a whole, where only the ratios between components are meaningful and the absolute scale does not carry any information (Aitchison, 1982). Such data are naturally represented on the interior simplex

$$\Delta^{K-1} := \left\{ \boldsymbol{z} \in \mathbb{R}_{>0}^K \;\middle|\; \sum_{k=1}^K z_k = 1 \right\}. \tag{1}$$

Equivalently, any two positive vectors in $\mathbb{R}_{>0}^K$ that differ only by a global scaling represent the same composition, since closure rescales them to sum to one. Consequently, valid analyses must be invariant to scale, respect relative information, and be coherent under subcompositions (choosing a subset of components and reclosing to the simplex). The canonical geometry for compositional data is the Aitchison geometry, which endows the simplex with a Euclidean structure via log-ratios (Aitchison, 1982); distances and operations then depend exclusively on relative relationships between components.

**Compositional Graph Representation Learning.** We consider graphs whose latent structure is compositional: each node $i$ is associated with a graded, overlapping mixture over $K$ latent archetypal factors. Concretely, we represent each node by a composition $\mathbf{z}_i \in \Delta^{K-1}$. Our inductive bias is that edge formation (i.e., the probability of an edge $(i, j) \in E$) is governed by the similarity of these relative mixtures, the ratios between archetypal weights, rather than by absolute latent scale or magnitude (e.g., the norm of an unconstrained embedding).

Accordingly, each node $i \in V$ is associated with a latent composition of $K$ components

$$\mathbf{z}_i = (z_{i1}, \ldots, z_{iK}) \in \Delta^{K-1}, \tag{2}$$

where each dimension $k$ corresponds to a latent archetypal factor, and $z_{ik}$ denotes the relative contribution of archetype $k$ to node $i$. The simplex $\Delta^{K-1}$ thus parameterizes nodes' roles as full compositions over archetypal factors.

Under this interpretation, simplex vertices correspond to idealized pure archetypes, in which a node is dominated by a single latent factor, while interior points represent graded, mixed roles. In many graphs, absolute interaction volume is a nuisance factor: for example, two nodes may differ greatly in degree or activity while exhibiting the same relative pattern of interactions across archetypal behaviors. When node roles are compositional, meaningful similarity should therefore depend on relative trade-offs between archetypes (e.g., how interactions are distributed across patterns) rather than absolute magnitudes (e.g., how many interactions occur). To respect these semantics, we represent nodes' roles as compositions and endow $\Delta^{K-1}$ with the Aitchison geometry, the canonical framework for comparing relative mixtures. This geometry is scale invariant and subcompositionally coherent, ensuring that restricting attention to any subset of archetypal factors preserves their relative relationships. As a result, latent similarity reflects relative archetypal dominance rather than overall activity, naturally supporting graded and overlapping node roles.

Our objective is to learn a latent geometry on the simplex that captures similarity between node roles in terms of relative archetypal composition. We do not assume that nodes correspond to pure archetypes; instead, roles are expressed as graded mixtures with overlapping structure. In the following section, we show how this geometry can be isometrically embedded in Euclidean space via log-ratio coordinates, enabling efficient optimization through unconstrained gradient-based methods in $\mathbb{R}^{K-1}$ while exactly preserving the Aitchison geometry of the simplex.

**Aitchison Geometry and ILR Coordinates** The simplex $\Delta^{K-1}$ endowed with the Aitchison geometry is not a Euclidean space: compositional observations are identifiable only up to a positive scaling, so only relative information (ratios between components) is meaningful. The Aitchison geometry therefore defines distances and inner products in terms of (log-)ratios, yielding a scale-invariant notion of similarity on $\Delta^{K-1}$. To enable efficient optimization using standard Euclidean tools while preserving this geometry, we embed compositions into $\mathbb{R}^{K-1}$ via the isometric log-ratio (ILR) transformation.

Let $\mathbf{1} \in \mathbb{R}^K$ denote the all-ones vector, and define the contrast space

$$\mathcal{C} = \left\{ \mathbf{v} \in \mathbb{R}^K \;\middle|\; \mathbf{v}^\top \mathbf{1} = 0 \right\}, \tag{3}$$

which contains all valid log-ratio contrasts between archetypal factors (see Appendix). The contrast space $\mathcal{C}$ is a $(K-1)$-dimensional linear subspace of $\mathbb{R}^K$; the ILR transform provides Euclidean coordinates for elements of $\mathcal{C}$ in $\mathbb{R}^{K-1}$. Let $\mathbf{V} \in \mathbb{R}^{K \times (K-1)}$ be any matrix whose columns form an orthonormal basis of $\mathcal{C}$, satisfying

$$\mathbf{V}^\top \mathbf{V} = \mathbf{I}, \qquad \mathbf{V}^\top \mathbf{1} = \mathbf{0}, \tag{4}$$

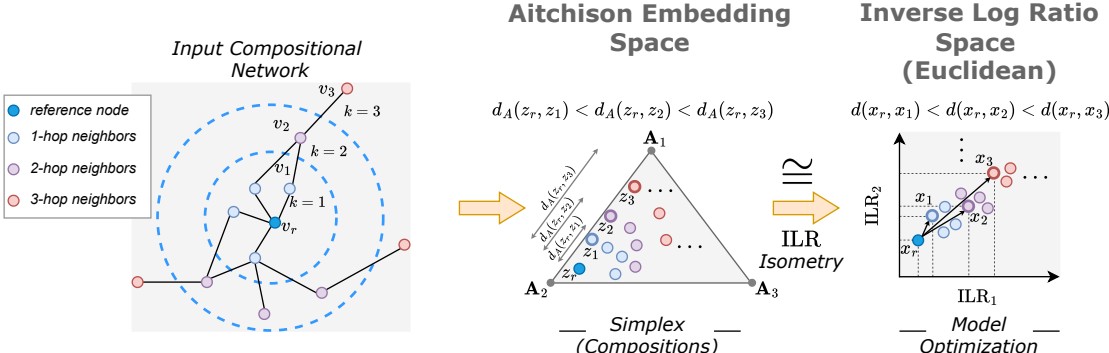

*Figure 1.* **Overview of AICoG.** Nodes are represented as compositions on the simplex and compared in Aitchison geometry. An ILR isometry maps compositions to an unconstrained Euclidean space, where distances preserve Aitchison distances.

where $\mathbf{I} \in \mathbb{R}^{(K-1)\times(K-1)}$ is the identity matrix. Given a node role $\mathbf{z}_i \in \Delta^{K-1}$, its ILR coordinates are defined as

$$\mathbf{x}_i = \text{ILR}(\mathbf{z}_i) = \log(\mathbf{z}_i)^\top \mathbf{V} \in \mathbb{R}^{K-1}, \qquad (5)$$

where the logarithm is applied elementwise. Each coordinate of $\mathbf{x}_i$ corresponds to a contrast between groups of latent archetypal factors, i.e., a log-ratio comparing their relative contributions, encoding relative dominance rather than absolute membership (Aitchison, 1982; Egozcue et al., 2003).

Importantly, the Aitchison distance between two compositions $\mathbf{z}_i, \mathbf{z}_j \in \Delta^{K-1}$ can now be defined as the Euclidean distance between their ILR representations (Aitchison, 1982):

$$d_A(\mathbf{z}_i, \mathbf{z}_j) = \|(\text{ILR}(\mathbf{z}_i) - \text{ILR}(\mathbf{z}_j))\|_2, \qquad (6)$$

A key property of the ILR transformation is that it is an isometry between $(\Delta^{K-1}, d_A)$ and $\mathbb{R}^{K-1}$, hence for $\mathbf{x}_i = \text{ILR}(\mathbf{z}_i)$,

$$d_A(\mathbf{z}_i, \mathbf{z}_j) = \|\mathbf{x}_i - \mathbf{x}_j\|_2. \qquad (7)$$

Consequently, Euclidean distances between ILR embeddings exactly preserve similarity in relative archetypal composition.

Log-ratio coordinates are not unique: the additive log-ratio (ALR), the centered log-ratio (CLR), and the isometric log-ratio (ILR) transform all map compositions into real-valued vector spaces (Greenacre et al., 2023). We prefer ILR because it provides an orthonormal coordinate system in $\mathbb{R}^{K-1}$ that is distance-preserving with respect to the Aitchison metric. In contrast, ALR depends on an arbitrary reference component and is not an isometry, while CLR produces coordinates in a constrained (sum-to-zero) subspace of $\mathbb{R}^K$. Any two valid ILR bases are related by an orthogonal transformation and therefore induce identical distances and likelihoods, decoupling geometric structure from coordinate representation.

**Lemma 3.1** (Subcomposition corresponds to a projection of ILR embeddings). *Let $z_i \in \Delta_\circ^{K-1}$ be compositional embeddings and let $S \subset \{1, \ldots, K\}$ with $|S| \geq 2$. Define the re-closed subcomposition*

$$z_i^{(S)} = \mathcal{C}\big((z_{i,r})_{r\in S}\big) \in \Delta_\circ^{|S|-1},$$

*where $\mathcal{C}$ denotes closure. Let $\text{ILR}$ and $\text{ILR}_S$ be ILR transforms on $\Delta_\circ^{K-1}$ and $\Delta_\circ^{|S|-1}$, respectively.*

*Then there exists a linear map $P_S \in \mathbb{R}^{(|S|-1)\times(k-1)}$ with orthonormal rows such that for all $i, j$,*

$$\big\| \text{ILR}_S(z_i^{(S)}) - \text{ILR}_S(z_j^{(S)}) \big\|_2 = \big\| P_S\big( \text{ILR}(z_i) - \text{ILR}(z_j)\big) \big\|_2.$$

**Node Representation.** To learn simplex-valued node roles while enabling unconstrained optimization for AICoG, we parameterize each composition $\mathbf{z}_i \in \Delta^{K-1}$ via an unconstrained vector (logits) $\tilde{\mathbf{z}}_i \in \mathbb{R}^K$ using a row-wise softmax transformation, $\mathbf{z}_i = \text{softmax}(\tilde{\mathbf{z}}_i)$. Given $\mathbf{z}_i \in \Delta^K$, we compute its ILR embedding $\mathbf{x}_i = \text{ILR}(\mathbf{z}_i)$ via Eq. (5).

**Edge likelihood.** We consider binary undirected graphs and model edges using a Bernoulli likelihood. Let $\mathbf{x}_i \in \mathbb{R}^{K-1}$ denote the ILR embedding of node $i$, and let $\gamma_i \in \mathbb{R}$ be a node-specific bias that captures degree heterogeneity.

For an unordered node pair $(i, j)$ with $i < j$, we define the log-odds

$$\eta_{ij} = -\|\mathbf{x}_i - \mathbf{x}_j\|_2 + \gamma_i + \gamma_j, \qquad (8)$$

, so that the Bernoulli edge probability is a monotone decreasing function of the ILR distance (equivalently, the Aitchison distance). This makes AICoG essentially a latent distance model (Hoff et al., 2002) under Aitchison geometry.

The model is trained by maximizing the Bernoulli log-likelihood over observed edges.

The complete log-likelihood over unordered pairs $\mathcal{D} = \{(i,j) : 1 \le i < j \le N\}$ is

$$\log p(\mathbf{Y} \mid \{\mathbf{X}\}, \{\boldsymbol{\gamma}\}) = \sum_{i<j} \left[ Y_{ij}\,\eta_{ij} - \log(1 + \exp(\eta_{ij})) \right]. \tag{9}$$

Importantly, the model in Eq. (8) has the same expressive power as an unconstrained Euclidean latent distance model in $\mathbb{R}^{K-1}$ (Theorem 3.2). Consequently, imposing a compositional inductive bias does not restrict the class of edge–probability matrices that can be represented in principle. Even when the assumption of compositional node roles is weak or absent, the proposed formulation is therefore no less expressive than a standard Euclidean latent distance model. Our goal is not to improve raw expressivity over Euclidean embeddings, but to endow latent distances with principled, invariant semantics when node roles are compositional, at no additional expressive cost.

**Theorem 3.2** (Expressive equivalence of ILR-compositional latent distance models)**.** *Let $K \ge 2$, let $g : [0,\infty) \to \mathbb{R}$, and let $\sigma : \mathbb{R} \to (0,1)$ be a link function. Let $\Delta_\circ^{K-1}$ denote the open simplex, and let $\mathrm{ILR} : \Delta_\circ^{K-1} \to \mathbb{R}^{K-1}$ be an isometric bijection.*

*Consider the latent distance model with latent positions $z_i \in \Delta_\circ^{K-1}$:*

$$\mathbb{P}(A_{ij} = 1) = \sigma\left(\alpha - g(\|\,\mathrm{ILR}(z_i) - \mathrm{ILR}(z_j)\,\|_2)\right).$$

*Then the set of edge–probability matrices realizable by this model is identical to that of the Euclidean latent distance model*

$$\mathbb{P}(A_{ij} = 1) = \sigma\left(\alpha - g(\|x_i - x_j\|_2)\right), \quad \mathbf{x}_i \in \mathbb{R}^{K-1}.$$

The ILR transformation depends on the choice of an orthonormal ILR basis $\mathbf{V}$ of the contrast space $\mathcal{C} = \{\mathbf{v} \in \mathbb{R}^K : \mathbf{v}^\top \mathbf{1} = 0\}$. While all valid bases induce identical Aitchison distances, different choices correspond to different systems of log-ratio contrasts among latent archetypal factors and therefore affect interpretability. A common default is the Helmert basis, which defines simple hierarchical contrasts and is domain-agnostic and computationally efficient. Alternatively, sequential binary partition (SBP) bases define contrasts via recursive binary splits, yielding interpretable balances when domain structure is known, but requiring a predefined partition. Importantly, to enable data-driven and informative coordinate system for human interpretation of compositional contrasts, we introduce a learnable ILR basis (see Appendix) that is trained jointly with the node embeddings. A key advantage of working in Aitchison geometry is that interpretability is a property of the representation space itself, rather than of a particular coordinate system. Any orthonormal ILR basis yields valid log-ratio contrasts; different bases simply correspond to different but equivalent views of the same compositional representation.

*Table 1.* Dataset statistics. $N$: number of nodes, $|E|$: number of edges, $K$: number of labels; "—" indicates unlabeled datasets.

|  | LastFM | Citeseer | Cora | Dblp | AstroPh | GrQc | HepTh |
|---|---|---|---|---|---|---|---|
| $N$ | 7,624 | 3,327 | 2,708 | 27,199 | 17,903 | 5,242 | 8,638 |
| $|E|$ | 55,612 | 9,104 | 5,278 | 66,832 | 197,031 | 14,496 | 24,827 |
| $K$ | 14 | 6 | 7 | — | — | — | — |

**Geometric roles and interpretability.** In our framework, a role is a pattern of interaction behavior captured by proximity in the learned latent geometry: nodes $i$ and $j$ occupy similar roles when their representations are close in Aitchison geometry, i.e., when $d_A(\mathbf{z}_i, \mathbf{z}_j)$ is small (equivalently $\|\mathbf{x}_i - \mathbf{x}_j\|_2$ is small in ILR space). Roles are therefore not associated with individual simplex components or coordinate axes; instead, they correspond to regions of the latent space containing nodes with similar relative archetypal compositions and interaction profiles. This yields continuous, overlapping roles that are invariant to orthogonal reparameterizations of the ILR coordinates, aligning with regular equivalence rather than cohesive communities. AICoG provides geometric and compositional interpretability rather than coordinate identifiability of a unique latent axis.

This interpretability appears at three levels. First, at the geometric level, distances correspond to differences in relative archetypal mixtures: two nodes are close when their log-ratio trade-offs among archetypal factors are similar. Second, under any chosen ILR basis, coordinates provide balance-level views, where each balance is a log-ratio contrast between groups of archetypes. These balances are not unique, since valid ILR bases are related by orthogonal transformations, but each basis yields a valid interpretable view of the same Aitchison geometry. Third, at the component level, the compositional representation supports subcompositional restriction: one can restrict attention to a subset of archetypes, re-close the composition, and analyze how predictions or separations change. Thus, interpretability is attached to the invariant compositional geometry and to valid log-ratio views of that geometry, rather than to a single privileged coordinate system.

This geometric definition also induces a distinct notion of explainability. Classical mixed-membership models (e.g., MMSBM) provide coordinate-level explanations in which each latent dimension is an identifiable role (up to permutation) and node vectors are membership weights along these axes. In contrast, because our likelihood depends only on distances in ILR space, the representation is orthogonally invariant and individual coordinates are not semantically identifiable; meaning is attributed to invariant geometric structure.

Unlike general Euclidean embeddings, where geometry is identifiable but lacks intrinsic semantic grounding, Aitchison geometry endows distances with a compositional in-

terpretation: they quantify differences in relative mixtures (log-ratio trade-offs) among latent archetypal factors. Operationally, each node is a composition $\mathbf{z}_i \in \Delta^{K-1}$ optimized via the ILR coordinates in $\mathbb{R}^{K-1}$; under a chosen ILR basis, coordinates correspond to balances (log-ratio contrasts), and subcompositional coherence ensures that the restriction to a subset of components and the application of closure remains well-defined (Lemma 3.1). Consequently, proximity and predicted links admit direct explanations in terms of stable relative trade-offs, rather than arbitrary coordinate assignments.

**Computational complexity.** Naively, evaluating the Bernoulli–logistic edge likelihood scales as $\mathcal{O}(N^2)$ due to the all-pairs distance-matrix computation. To achieve scalability, we note that the first term of Eq. (9) depends only on the observed edges and can be computed in $\mathcal{O}(|E|)$ time. The log-partition term is approximated via uniform subsampling of non-edges, yielding an unbiased estimator. With a number of samples proportional to $|E|$, this reduces the per-iteration complexity to $\mathcal{O}(|E|)$.

# 4. Results

We extensively evaluate AICoG against prominent baseline graph representation learning methods, including both unconstrained and simplex-based representations, across networks of varying sizes and structures. We ran all methods using publicly available implementations (see Appendix). When GPU support was available, experiments were executed on an NVIDIA A100 GPU; otherwise, we used an Apple M2 machine with 8 GB RAM. For AICoG, we optimize the Bernoulli negative log-likelihood in Eq. (9) using Adam (Kingma & Ba, 2014) (learning rate $10^{-2}$) for 5,000 iterations. Unless stated otherwise, the primary tuned hyperparameter is the embedding dimension $D$; for simplex-based methods with $K$ components, we report $D = K - 1$ (the corresponding ILR dimension), and use the same $D$ convention for all methods. Proofs regarding theorems and lemmas are provided in the Appendix. The code is provided as supplementary material and will be released publicly.

**Datasets and Baselines.** We evaluate on citation, coauthorship, and social graphs, where edges are plausibly driven by multiple latent factors (e.g., topical affinity, methodology, or social proximity) rather than a single categorical role. Accordingly, we represent each node as a mixture over latent archetypes and learn embeddings that capture how these archetypal influences shape connectivity (Table 1). We consider the citation networks *Cora* and *Citeseer* (Sen et al., 2008), the coauthorship networks *DBLP*, *AstroPh*, *GrQc*, and *HepTh* (Perozzi et al., 2017; Leskovec et al., 2007), and the social network *LastFM* (Rozemberczki & Sarkar, 2020). We compare against (i) Euclidean vector-space embedding methods, (ii) matrix factorization methods, and

(iii) mixed-membership / simplex-based models. Euclidean vector-space embeddings include NODE2VEC (Grover & Leskovec, 2016) and ROLE2VEC (Ahmed et al., 2018), which learn node representations in $\mathbb{R}^d$ and rely on Euclidean inner-product geometry. Factorization-based methods include NETMF (Qiu et al., 2018). Mixed-membership baselines include MMSBM (Airoldi et al., 2007), and MNMF (Wang et al., 2017). Latent distance Simplex-based baselines include SLIM-RAA (Nakis et al., 2023a) and HM-LDM (Nakis et al., 2022). Finally, we include SIMPLEX-EUCLIDEAN, a latent distance model operating directly on the simplex with Euclidean geometry (without an ILR/Aitchison transformation), to isolate the effect of compositional geometry. We focus on featureless graphs and unsupervised representation learning, a regime in which message-passing GNNs are known to perform poorly without auxiliary features or supervision (Nikolentzos et al., 2024) and thus are omitted.

**Link prediction.** We follow the standard link prediction protocol of Perozzi et al. (2014); Nakis et al. (2023b). For each dataset, we uniformly remove $50\%$ of the edges while ensuring that the remaining graph is connected. The removed edges, together with an equal number of randomly sampled non-edges, form the test set; embeddings are learned on the residual graph only. We evaluate on five benchmark networks over five random runs and multiple embedding dimensions $D \in \{8, 16, 32, 64\}$. Performance is reported as AUC-ROC in Table 2 (PR-AUC is deferred to the Appendix). Following Grover & Leskovec (2016), for embedding methods we construct dyadic features using standard binary operators (average, Hadamard, weighted-$L_1$, weighted-$L_2$) and train an $L_2$-regularized logistic regression classifier. For likelihood-based models, we instead compute link probabilities directly from the learned log-odds $\eta_{ij}$, without an auxiliary classifier. We report three variants of our method: AICoG (HB) (fixed Helmert ILR basis), AICoG (LB) (learned ILR basis), and AICoG (HB) SUBCOMP, which evaluates subcompositional restriction of a trained model (details in Appendix).

Results show that AICoG achieves on-par or favorable predictive performance relative to the strongest baselines. Replacing Aitchison geometry with Euclidean geometry on the simplex (SIMPLEX-EUCLIDEAN) leads to a substantial performance drop, indicating that compositional geometry, rather than simplex constraints alone, is critical. Among mixed-membership baselines, MMSBM and MNMF underperform across most datasets. The strongest competitors are SLIM-RAA and HM-LDM, which increase expressivity via linear expansions of the simplex and occasionally achieve comparable performance. The subcompositional evaluation shows that removing components (with closure) preserves downstream utility under substantial compression, and performance is stable across different ILR bases,

*Table 2.* AUC ROC scores for representation sizes of 8, 16, 32, and 64 averaged over five runs.

| | AstroPh | | | | GrQc | | | | HepTh | | | | Cora | | | | DBLP | | | |
|---|---|---|---|---|---|---|---|---|---|---|---|---|---|---|---|---|---|---|---|---|
| Dimension ($D$) | 8 | 16 | 32 | 64 | 8 | 16 | 32 | 64 | 8 | 16 | 32 | 64 | 8 | 16 | 32 | 64 | 8 | 16 | 32 | 64 |
| NODE2VEC | .943 | .954 | .961 | .962 | .928 | .932 | .937 | .936 | .879 | .882 | .888 | .892 | .761 | .760 | .766 | .777 | .920 | .923 | .931 | .941 |
| ROLE2VEC | .957 | .969 | .970 | .965 | .927 | .936 | .934 | .934 | .897 | .907 | .902 | .895 | .769 | .767 | .759 | .752 | .940 | .952 | .943 | .944 |
| NETMF | .904 | .928 | .946 | .955 | .835 | .882 | .882 | .883 | .778 | .797 | .802 | .793 | .698 | .675 | .674 | .654 | .791 | .817 | .829 | .842 |
| SLIM-RAA | **.965** | .970 | .969 | .969 | .940 | .943 | .947 | .949 | .902 | .914 | .919 | .920 | .797 | .782 | .784 | .789 | **.953** | **.959** | **.961** | .962 |
| MMSBM | .886 | .897 | .907 | .913 | .818 | .814 | .806 | .817 | .773 | .756 | .751 | .757 | .667 | .679 | .667 | .659 | .779 | .776 | .780 | .777 |
| MNMF | .877 | .916 | .938 | .954 | .881 | .905 | .918 | .918 | .810 | .844 | .864 | .875 | .694 | .718 | .700 | .680 | .859 | .898 | .919 | .931 |
| HM-LDM | .940 | .941 | .944 | .950 | .937 | .939 | .940 | .944 | .876 | .876 | .882 | .886 | .808 | .809 | .800 | .806 | .890 | .876 | .891 | .921 |
| SIMPLEX-EUCLIDEAN | .853 | .848 | .849 | .863 | .831 | .817 | .808 | .808 | .750 | .738 | .737 | .751 | .739 | .723 | .718 | .709 | .694 | .678 | .691 | .737 |
| AICoG (HB) SUBCOMP | .962 | .970 | .974 | **.976** | .948 | .955 | .959 | **.961** | .905 | .918 | .926 | **.929** | .807 | .829 | .849 | .851 | .946 | .956 | **.961** | **.963** |
| AICoG (HB) | **.965** | .972 | .975 | .976 | .953 | .958 | .961 | .961 | .911 | **.927** | .928 | **.929** | .837 | .846 | .850 | .851 | .949 | .958 | **.961** | .962 |
| AICoG (LB) | **.965** | **.973** | **.975** | **.976** | .953 | .959 | **.961** | .961 | **.911** | .926 | **.929** | .928 | .839 | .847 | .850 | **.852** | .949 | .958 | **.961** | **.963** |

*Table 3.* Micro-F1 scores for representation sizes of 8, 16, 32, and 64 averaged over five runs.

| | Cora | | | | Citeseer | | | | LastFM | | | |
|---|---|---|---|---|---|---|---|---|---|---|---|---|
| Dimension ($D$) | 8 | 16 | 32 | 64 | 8 | 16 | 32 | 64 | 8 | 16 | 32 | 64 |
| NODE2VEC | .771 | .778 | .796 | .814 | .610 | .658 | .669 | .695 | .842 | .859 | .862 | .865 |
| ROLE2VEC | **.783** | .796 | .799 | .803 | **.677** | .693 | .694 | .706 | .837 | .853 | .857 | .854 |
| NETMF | .733 | .761 | .785 | .799 | .612 | .646 | .671 | .696 | .744 | .803 | .827 | .837 |
| SLIM-RAA | .628 | .632 | .684 | .729 | .564 | .583 | .580 | .630 | .697 | .777 | .807 | .822 |
| MMSBM | .313 | .406 | .534 | .614 | .393 | .464 | .541 | .567 | .259 | .405 | .499 | .673 |
| MNMF | .516 | .612 | .678 | .722 | .457 | .506 | .571 | .622 | .661 | .734 | .780 | .796 |
| HM-LDM | .697 | .718 | .777 | .814 | .620 | .613 | .655 | .691 | .799 | .801 | .817 | .832 |
| SIMPLEX-EUCLIDEAN | .442 | .457 | .485 | .442 | .454 | .404 | .414 | .431 | .440 | .455 | .478 | .548 |
| AICoG (HB) SUBCOMP | .672 | .780 | .821 | .831 | .577 | .668 | .713 | .736 | .795 | .857 | .795 | .870 |
| AICoG (HB) | .769 | **.806** | **.829** | .831 | .671 | **.702** | **.721** | .736 | .853 | .867 | .869 | **.870** |
| AICoG (LB) | .780 | .804 | .826 | **.833** | .674 | .704 | .717 | .733 | .853 | .868 | .870 | **.870** |

*Table 4.* Synthetic membership recovery. Lower $\ell_1$ and JS divergence are better; higher cosine similarity is better. AICoG is closer to the true memberships across all regimes.

| Generator / regime | Model | $\ell_1 \downarrow$ | Cosine $\uparrow$ | JS $\downarrow$ |
|---|---|---|---|---|
| Bilinear-continuous | AICoG | 0.922 | 0.624 | 0.161 |
| | MMSBM | 1.456 | 0.423 | 0.360 |
| Bilinear-discrete | AICoG | 1.082 | 0.619 | 0.257 |
| | MMSBM | 1.617 | 0.223 | 0.513 |
| ILR-continuous | AICoG | 0.900 | 0.645 | 0.154 |
| | MMSBM | 1.452 | 0.432 | 0.356 |
| ILR-discrete | AICoG | 1.383 | 0.372 | 0.407 |
| | MMSBM | 1.581 | 0.244 | 0.501 |

consistent with the orthogonal invariance of the Aitchison representation.

**Node classification.** We follow the standard node classification protocol of (Perozzi et al., 2014) on labeled networks, using node embeddings as fixed features for a multinomial logistic regression classifier. Labeled nodes are split into training, validation, and test sets, with validation used to tune regularization. Table 3 reports average Micro-F1 scores over five runs using identical splits across methods (macro-F1 in the Appendix). AICoG achieves favorable or on-par performance relative to all baselines. As expected, Euclidean embeddings yield the strongest overall accuracy, while mixed-membership and simplex-based methods perform worse, with SLIM-RAA being the most competitive among them. Overall, AICoG maintains competitive predictive performance while yielding geometrically interpretable role representations by construction. Performance is stable across ILR bases, and subcompositional restriction preserves downstream utility without retraining.

**Synthetic membership recovery.** We also evaluate AICoG in a controlled synthetic setting where the true memberships and edge-generation mechanism are known. We cross two membership regimes, continuous/interior and near-discrete, with two edge generators: an MMSBM-style bilinear model and an ILR-distance model. This separates the effect of the latent membership regime from the edge model. We compare learned memberships to the ground truth using $\ell_1$ distance, cosine similarity, and Jensen–

Shannon divergence. As summarized in Table 4, AICoG recovers memberships closer to the truth than MMSBM across all four settings, with the largest gains in the continuous/interior regimes. MMSBM tends to collapse toward near-discrete assignments even when the true memberships are interior, while AICoG remains closer to the underlying compositional structure.

**Membership interiority and overlap.** To test whether AICoG learns genuinely overlapping simplex-valued roles rather than near-discrete assignments, we compare the learned memberships of AICoG against MMSBM on Cora. We report four statistics: entropy, where larger values indicate more overlap; maximum component mass, where larger values indicate more nearly one-hot assignments; near-corner fraction, the percentage of nodes whose largest component exceeds a high threshold; and the effective number of active roles, defined as $\exp(H(z_i))$ and averaged over nodes. Table 5 shows that AICoG learns substantially more interior memberships than MMSBM. Moreover, a classifier trained directly on the learned memberships achieves much higher accuracy and macro-F1 for AICoG, indicating that this overlap is label-informative rather than diffuse noise.

**Geometric Explainability through Compositional Balances.** Figure 2 shows label-wise distributions along indi-

*Table 5.* Membership interiority on Cora. AICoG learns substantially more overlapping and interior compositions than MMSBM. The membership-probe results show that this additional overlap is label-informative.

| Model | Entropy ↑ | Max comp. ↓ | Near-corner ↓ | Eff. roles ↑ | Probe Acc. ↑ | Probe Macro-F1 ↑ |
|---|---|---|---|---|---|---|
| AICoG | 1.064 | 0.603 | 5.55% | 3.07 | 0.670 | 0.657 |
| MMSBM | 0.191 | 0.935 | 78.95% | 1.24 | 0.306 | 0.121 |

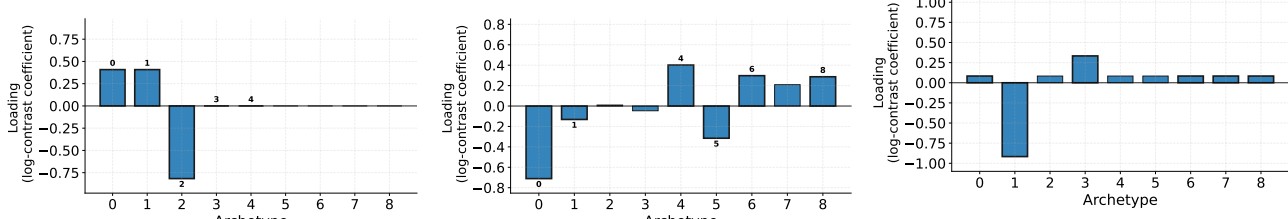

*(a)* Helmert basis: balance loadings (coord 1). *(b)* Learned basis: balance loadings (coord 3). *(c)* Varimax learned basis: balance loadings (coord 7).

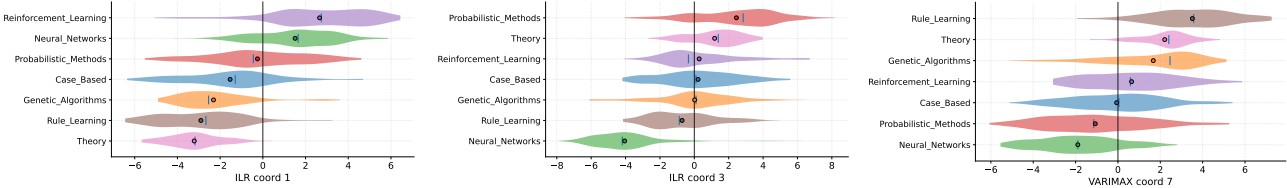

*(d)* Helmert basis: label-wise distributions along the selected balance. *(e)* Learned basis: label-wise distributions along the selected balance. *(f)* Varimax learned basis: label-wise distributions along the selected balance.

*Figure 2.* **Cora dataset** ($D=8$). Label-wise distributions along ILR balances under three valid bases: Helmert (left), learned (center), and varimax-rotated learned (right). Top row shows balance loadings (archetypal contributions to each log-ratio contrast); bottom row shows label-wise distributions of the corresponding ILR coordinates.

vidual ILR coordinates under three valid ILR bases: a fixed Helmert basis, a learned basis, and a varimax rotation of the learned basis. Each ILR coordinate corresponds to a log-ratio balance of the form $x_{ib} = \langle \log(\mathbf{z}_i), \mathbf{v}_b \rangle$, where $\mathbf{v}_b$ defines a contrast between groups of archetypal components. Although the specific balances differ across bases, as expected under orthogonal reparameterizations, label separation is consistent across all views, indicating that it reflects invariant structure of the underlying Aitchison geometry rather than a particular coordinate choice. The Helmert basis provides a neutral reference, the learned basis aligns separation with data-adapted directions, and the varimax rotation yields sparser, more interpretable balances. Crucially, these visualizations admit a compositional explanation: labels differ in stable log-ratio trade-offs among shared archetypal factors. This form of explanation is unavailable to unconstrained Euclidean embeddings, which rely on abstract directions, and to mixed-membership models, which assume axis-aligned roles. AICoG instead provides a continuous, geometry-level explanation in terms of relative compositions, without requiring identifiable roles.

We further quantify whether individual compositional balances capture label-relevant structure. For each learned ILR basis, a balance has the form $x_{ib} = \langle \log z_i, v_b \rangle$, corresponding to a log-ratio contrast between groups of archetypes. We

identify the most label-associated balance and evaluate it using three one-dimensional statistics: a 1D logistic probe trained using only that balance, an ANOVA $F$-statistic measuring between-label separation relative to within-label variation, and mutual information with the node label. On Cora, a single learned balance achieves approximately $0.40$ 1D probe accuracy, ANOVA $F \approx 319$, and mutual information $\approx 0.44$. This shows that individual log-ratio trade-offs can isolate meaningful label-associated structure, rather than interpretability arising only from the full high-dimensional embedding.

**Interpretable trade-off trajectories.** To illustrate geometric explainability in the learned compositional latent space, we define a controlled intervention directly on the simplex. For a node with composition $\mathbf{z} \in \Delta^{K-1}$ and two archetypal components $(a, b)$, we construct a paired log-ratio trajectory by reweighting these components in opposite directions, followed by closure: $z_a(s) \propto z_a e^s$, $z_b(s) \propto z_b e^{-s}$ and $z_k(s) \propto z_k$ $(k \notin \{a, b\})$, $\mathbf{z}(s) = \mathcal{C}(\mathbf{z}(s))$, where $s \in \mathbb{R}$ controls the trade-off strength and. This produces a smooth increase of the archetype $a$ compared to $b$ while preserving compositional constraints. We map $\mathbf{z}(s)$ to the ILR coordinates and visualize the resulting curve using PCA (for visualization only), in Figure 3. Crucially, the trajectory is defined intrinsically on the simplex and is therefore invari-

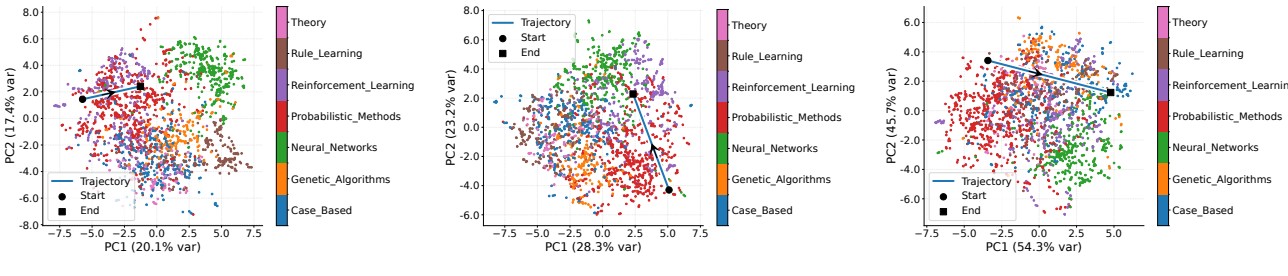

*(a) $K{=}9$ (full composition).*    *(b) $K{=}6$ subcomposition (renormalized).*    *(c) $K{=}3$ subcomposition (renormalized).*

*Figure 3.* **Interpretable trade-off trajectories under subcomposition (Cora).** Node embeddings are shown in a 2D PCA projection of ILR coordinates (Helmert basis), with PCA fit separately for each $K$ and nodes colored by label. The overlaid curve traces a paired log-ratio intervention applied to the same node across panels, increasing archetype $a$ and decreasing $b$ followed by closure. The intervention is defined intrinsically on the simplex, illustrating an interpretable trajectory under principled subcomposition.

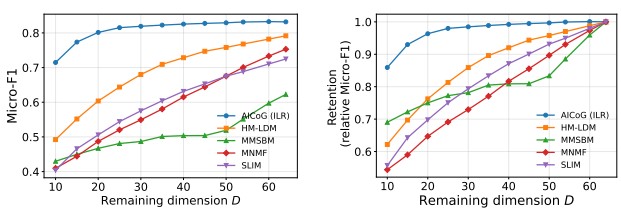

*(a)* **Absolute performance.** Micro-F1 after component removal.    *(b)* **Retention.** Relative utility retained compared to the full model.

*Figure 4.* **Subcompositional evaluation on CORA (trained at $D{=}64$).** We evaluate semantically meaningful component removal by restricting each simplex-based representation to $K'$ components, applying closure, and probing the resulting $D{=}K'-1$ embeddings without retraining. Curves are averaged over 50 random removal masks; higher is better in both panels.

ant to the choice of ILR basis: different bases yield different balance-based views of the same underlying trade-off geometry. Such semantics-preserving interventions are not available in unconstrained Euclidean embeddings, where coordinate changes are arbitrary, nor in mixed-membership models, which only support axis-aligned interpolation between discrete roles. The ability to construct smooth, basis-invariant trade-off paths is a direct consequence of equipping the latent simplex with Aitchison geometry.

**Subcompositional evaluation.** We evaluate semantically meaningful component removal by restricting each learned composition to a subset of components and applying closure, then measuring downstream utility as the number of retained components decreases (no retraining). We exclude unconstrained Euclidean baselines from this comparison because they do not admit an analogous operation: Euclidean coordinates are basis-dependent and lack a semantic interpretation as components, so "dimension dropping" is necessarily heuristic, and its apparent robustness often reflects representational redundancy rather than semantic stability. Results are reported for $D{=}64$ em-

beddings on CORA in Figure 4: panel (a) shows absolute Micro-F1 as a function of the remaining dimension, while panel (b) reports retention relative to the full model, $\mathrm{retain}(K') = \mathrm{Micro-F1}(K')/\mathrm{Micro-F1}(K_{\mathrm{full}})$. For each simplex-based method, we remove components to obtain $K' = K_{\mathrm{full}} - K_{\mathrm{rem}}$, apply closure, and evaluate the resulting $D = K' - 1$ dimensional representations; all curves are averaged over 50 random removal masks. We observe that AI-COG exhibits smooth degradation and achieves the strongest retention under component removal among simplex-based baselines, with the advantage most pronounced under aggressive compression.

## 5. Conclusion & Limitations

We introduced Aitchison Compositional Graph embeddings (AICoG), which represents nodes as compositions over latent archetypal factors and models edges via distances in Aitchison geometry. Using an ILR isometry, AICoG retains the expressivity of Euclidean latent distance models while grounding similarity in relative trade-offs and enabling semantically meaningful component restriction. Empirically, AICoG is competitive in link prediction and node classification, showing that imposing compositional semantics can produce interpretable representations without sacrificing predictive performance in our benchmarks.

The approach is most appropriate when node roles are naturally compositional; outside this regime, it may not improve accuracy over unconstrained Euclidean embeddings. Our experiments primarily follow standard connected-graph protocols, where the training graph is connected or dominated by a large connected component. An interesting direction is to extend AICoG to disconnected graphs or graphs with many small connected components. In such settings, subcompositional structure may provide a principled way to analyze whether different components rely on partially independent subsets of archetypal factors, while preserving coherent Aitchison geometry within each subcomposition.

## ACKNOWLEDGEMENTS

We gratefully acknowledge the reviewers for their constructive feedback and insightful comments. We also thank the area chair for their supportive meta-review and valuable suggestions for future research directions, which helped further strengthen our paper. N.N. is supported by the NOMIS and Stavros Niarchos Foundation. C.K. is supported by the IdAML Chair hosted at ENS Paris-Saclay, Université Paris-Saclay.

## Impact Statement

Many real-world networks, including social, biological, and economic systems, exhibit role structures that are continuous, overlapping, and context-dependent. Such settings challenge common graph representation learning methods, which often struggle to encode and interpret roles in a principled way. Widely used Euclidean graph embeddings typically yield explanations that depend on arbitrary coordinate choices, while mixed-membership models, though allowing overlap, rely on restrictive assumptions about discrete, axis-aligned roles. By grounding latent graph representations in Aitchison geometry, a canonical framework for compositional data, we introduce Aitchison Compositional Graph embeddings (AICoG), which provide interpretable representations by construction. Our approach enables explanations based on relative trade-offs between latent factors, supports continuous and overlapping role structure, and allows principled reasoning about component influence and stability through semantically meaningful restriction. Beyond the specific model studied here, Aitchison geometry provides a general foundation for extending compositional inductive biases to other graph learning architectures, including message-passing neural networks operating on compositional node states. As a result, AICoG contributes to interpretable graph representation learning in domains where roles cannot be cleanly separated, broadening the applicability of explainable graph models to complex real-world networks without introducing additional modeling assumptions.

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

# Appendix

This appendix provides supplementary material supporting the main paper. We include formal proofs and derivations, details on the ILR parameterization, complete experimental settings and hyperparameters, and additional empirical results. These materials clarify the methodology and enable full reproducibility of the reported experiments.

## A. Proofs and Theoretical Results

**Lemma .1** (Subcomposition corresponds to a projection of ILR embeddings). *Let $z_i \in \Delta_\circ^{k-1}$ be compositional embeddings and let $S \subset \{1, \ldots, k\}$ with $|S| \geq 2$. Define the reclosed subcomposition*

$$z_i^{(S)} = \mathcal{C}\big((z_{i,r})_{r \in S}\big) \in \Delta_\circ^{|S|-1},$$

*where $\mathcal{C}$ denotes closure. Let $\mathrm{ILR}$ and $\mathrm{ILR}_S$ be ILR transforms on $\Delta_\circ^{k-1}$ and $\Delta_\circ^{|S|-1}$, respectively.*

*Then there exists a linear map $P_S \in \mathbb{R}^{(|S|-1) \times (k-1)}$ with orthonormal rows such that for all $i, j$,*

$$\big\| \mathrm{ILR}_S(z_i^{(S)}) - \mathrm{ILR}_S(z_j^{(S)}) \big\|_2 = \big\| P_S\big( \mathrm{ILR}(z_i) - \mathrm{ILR}(z_j) \big) \big\|_2.$$

*Proof.* Recall $z_i \in \Delta_\circ^{k-1} \subset \mathbb{R}^k$ is a vector such that $(z_i)_r > 0$ for all $r$ and $\mathbf{1}_k^\top z_i = 1$. Let $\log(z_i) \in \mathbb{R}^k$ denote the elementwise logarithm.

Let $V \in \mathbb{R}^{k \times (k-1)}$ be an ILR basis matrix with orthonormal columns spanning the contrast space, i.e.,

$$V^\top V = I_{k-1}, \qquad V^\top \mathbf{1}_k = 0.$$

The ILR coordinates of $z_i$ are the $(k-1)$-dimensional row vector

$$\mathrm{ILR}(z_i) \triangleq \log(z_i)^\top V \in \mathbb{R}^{k-1}.$$

Fix a subset $S \subseteq \{1, \ldots, k\}$ with $|S| = k' \geq 2$. Let $R_S \in \mathbb{R}^{k' \times k}$ be the coordinate-selection matrix that extracts the entries in $S$, so that $R_S z = (z_r)_{r \in S} \in \mathbb{R}^{k'}$ for any $z \in \mathbb{R}^k$. Define the closure (re-normalization) map $C : \mathbb{R}_{>0}^{k'} \to \Delta_\circ^{k'-1}$ by

$$C(u) \triangleq \frac{u}{\mathbf{1}_{k'}^\top u}.$$

The reclosed subcomposition of $z_i$ on $S$ is

$$z_i^{(S)} \triangleq C(R_S z_i) \in \Delta_\circ^{k'-1}.$$

Let $V_S \in \mathbb{R}^{k' \times (k'-1)}$ be an ILR basis matrix for $\Delta_\circ^{k'-1}$ satisfying

$$V_S^\top V_S = I_{k'-1}, \qquad V_S^\top \mathbf{1}_{k'} = 0,$$

and define the ILR coordinates of $z_i^{(S)}$ by

$$\mathrm{ILR}_S(z_i^{(S)}) \triangleq \log(z_i^{(S)})^\top V_S \in \mathbb{R}^{k'-1}.$$

Write $u_i \triangleq R_S z_i \in \mathbb{R}_{>0}^{k'}$ and $s_i \triangleq \mathbf{1}_{k'}^\top u_i > 0$. By definition of closure, $z_i^{(S)} = u_i / s_i$, hence

$$\log z_i^{(S)} = \log u_i - (\log s_i) \mathbf{1}_{k'}.$$

Multiplying by $V_S^\top$ and using $V_S^\top \mathbf{1}_{k'} = 0$ gives

$$\mathrm{ILR}_S(z_i^{(S)}) = V_S^\top \log z_i^{(S)} = V_S^\top \log u_i = V_S^\top \log(R_S z_i).$$

Therefore,

$$\mathrm{ILR}_S(z_i^{(S)}) - \mathrm{ILR}_S(z_j^{(S)}) = V_S^\top \big( \log(R_S z_i) - \log(R_S z_j) \big). \tag{10}$$

Define the padded matrix $\widetilde{V}_S \triangleq R_S^\top V_S \in \mathbb{R}^{k \times (k'-1)}$. Then

$$V_S^\top \log(R_S z) = (R_S^\top V_S)^\top \log z = \widetilde{V}_S^\top \log z,$$

so (10) becomes

$$\mathrm{ILR}_S(z_i^{(S)}) - \mathrm{ILR}_S(z_j^{(S)}) = \widetilde{V}_S^\top \big( \log z_i - \log z_j \big). \tag{11}$$

Moreover,

$$\widetilde{V}_S^\top \widetilde{V}_S = V_S^\top R_S R_S^\top V_S = V_S^\top I_{k'} V_S = I_{k'-1},$$

so the columns of $\widetilde{V}_S$ are orthonormal, and

$$\widetilde{V}_S^\top \mathbf{1}_k = V_S^\top R_S \mathbf{1}_k = V_S^\top \mathbf{1}_{k'} = 0,$$

so they lie in the contrast space $\{x \in \mathbb{R}^k : \mathbf{1}_k^\top x = 0\}$. Since the columns of $V$ form an orthonormal basis for this contrast space, there exists $A \in \mathbb{R}^{(k-1)\times(k'-1)}$ such that $\widetilde{V}_S = VA$; take $A \triangleq V^\top \widetilde{V}_S$. Then

$$A^\top A = \widetilde{V}_S^\top V V^\top \widetilde{V}_S = \widetilde{V}_S^\top \widetilde{V}_S = I_{k'-1},$$

so $A$ has orthonormal columns. Substituting $\widetilde{V}_S = VA$ into (11) gives

$$\mathrm{ILR}_S(z_i^{(S)}) - \mathrm{ILR}_S(z_j^{(S)}) = A^\top V^\top \big( \log z_i - \log z_j \big) =$$

$$= A^\top \big( \mathrm{ILR}(z_i) - \mathrm{ILR}(z_j) \big).$$

Define $P_S \triangleq A^\top \in \mathbb{R}^{(k'-1)\times(k-1)}$. Then $P_S P_S^\top = A^\top A = I_{k'-1}$, and taking Euclidean norms yields the claimed identity. Finally,

$$P_S = A^\top = (V^\top \widetilde{V}_S)^\top = \widetilde{V}_S^\top V = V_S^\top R_S V,$$

as stated. $\qquad \square$

**Theorem .2** (Expressive equivalence of ILR-compositional latent distance models). *Let $k \geq 2$, let $g : [0, \infty) \to \mathbb{R}$, and let $\sigma : \mathbb{R} \to (0, 1)$ be a link function. Let $\Delta_\circ^{k-1}$ denote the open simplex and let $\mathrm{ILR} : \Delta_\circ^{k-1} \to \mathbb{R}^{k-1}$ be an isometric bijection.*

*Consider the latent distance model*

$$\mathbb{P}(A_{ij} = 1) = \sigma(\alpha - g(\| \mathrm{ILR}(z_i) - \mathrm{ILR}(z_j) \|_2)), \, z_i \in \Delta_\circ^{k-1}.$$

*Then the set of edge–probability matrices realizable by this model is identical to that of the Euclidean latent distance model*

$$\mathbb{P}(A_{ij} = 1) = \sigma(\alpha - g(\|x_i - x_j\|_2)), \qquad x_i \in \mathbb{R}^{k-1}.$$

*Proof.* Since ILR is bijective, for any collection $\{x_i\}_{i=1}^n \subset \mathbb{R}^{k-1}$ there exists a unique collection $\{z_i\}_{i=1}^n \subset \Delta_\circ^{k-1}$ such that $x_i = \mathrm{ILR}(z_i)$ for all $i$, and conversely $z_i = \mathrm{ILR}^{-1}(x_i)$ exists for all $i$. Because ILR is an isometry, for all $i < j$,

$$\|x_i - x_j\|_2 = \| \mathrm{ILR}(z_i) - \mathrm{ILR}(z_j) \|_2.$$

Substituting this identity into the respective expressions for the edge probabilities shows that both parameterizations induce identical probabilities for every node pair. Therefore both models realize the same set of edge–probability matrices. $\quad \square$

## B. Model Parameterization and Interpretability

**Constrast space contains all valid log-ratio contrasts**  A linear log-contrast on $\mathbf{z} \in \mathbb{R}_{>0}^K$ can be written as $\mathbf{v}^\top \log \mathbf{z}$ for some $\mathbf{v} \in \mathbb{R}^K$ (elementwise log). Requiring invariance to positive rescaling, $\mathbf{z} \mapsto c\mathbf{z}$ for any $c > 0$, yields

$$\mathbf{v}^\top \log(c\mathbf{z}) = \mathbf{v}^\top \log \mathbf{z} \quad \forall \mathbf{z}, \, \forall c > 0.$$

Using $\log(c\mathbf{z}) = \log \mathbf{z} + (\log c)\mathbf{1}$, we obtain

$$\mathbf{v}^\top \log(c\mathbf{z}) = \mathbf{v}^\top \log \mathbf{z} + (\log c)\mathbf{v}^\top \mathbf{1}.$$

Thus the invariance condition holds for all $c > 0$ if and only if $\mathbf{v}^\top \mathbf{1} = 0$, i.e., $\mathbf{v} \in \mathcal{C} = \{\mathbf{v} \in \mathbb{R}^K : \mathbf{v}^\top \mathbf{1} = 0\}$. In particular, any pairwise log-ratio $\log(z_a/z_b) = \log z_a - \log z_b$ corresponds to $\mathbf{v} = \mathbf{e}_a - \mathbf{e}_b \in \mathcal{C}$, and more general group log-ratio contrasts are obtained by any $\mathbf{v} \in \mathcal{C}$ with positive weights on one group and negative weights on another (summing to zero).

**Learning a ILR basis.** Importantly, to enable data-driven and informative coordinate system for human interpretation of compositional contrasts, we introduce a learnable ILR basis that is trained jointly with the node embeddings. We parameterize an unconstrained matrix as, Let $W \in \mathbb{R}^{K \times (K-1)}$ be any unconstrained parameter matrix. Define

$$\widetilde{W} = W - \frac{1}{K} \mathbf{1}\mathbf{1}^\top W, \qquad V = \text{QR}(\widetilde{W}),$$

and take the first $K-1$ columns of $V$. Then $V$ is a valid ILR basis: (i) $V^\top V = I$ (orthonormality), (ii) $V^\top \mathbf{1} = 0$ (contrast constraint). Thus, any learned $W$ induces a valid ILR coordinate system.

**Geometric role definition.** In our framework, a *role* denotes a pattern of interaction behavior captured by proximity in the learned latent geometry. Formally, nodes $i$ and $j$ are said to occupy similar roles when their latent representations are close in Aitchison geometry, i.e., when $d_A(\mathbf{z}_i, \mathbf{z}_j)$ is small, equivalently when their ILR embeddings satisfy $\|\mathbf{y}_i - \mathbf{y}_j\|_2$ is small. Roles are not associated with individual simplex components or coordinate axes; instead, they correspond to regions of the latent space containing nodes with similar relative archetypal compositions and interaction profiles. Consequently, roles are continuous, overlapping, and invariant to orthogonal reparameterizations of the ILR coordinates, aligning with regular equivalence rather than cohesive communities.

**Geometric role explainability.** This geometric role definition leads to a distinct notion of explainability. Classical mixed-membership models, such as MMSBM, explain graph structure through *coordinate-level roles*: each latent dimension corresponds to an identifiable role (up to permutation), and node representations encode membership weights along these axes. In contrast, because our likelihood depends only on distances in ILR space, the representation is invariant under orthogonal reparameterizations, and individual coordinates or simplex components cannot be interpreted as roles. Instead, semantic meaning is attributed to invariant geometric structure. Unlike general Euclidean embeddings, where geometry is identifiable but lacks intrinsic semantic grounding, our model endows the latent space with Aitchison geometry, so distances correspond to differences in relative mixtures and principled operations such as subcomposition preserve meaning. As a result, roles in our framework are explainable by construction: their meaning is given by stable relative trade-offs among latent archetypal factors, rather than by arbitrary coordinate assignments.

**Operational interpretability.** Interpretation is built into the representation rather than added post hoc. Each node is embedded as a composition $\mathbf{z}_i \in \Delta^{K-1}$, which summarizes its relative emphasis over $K$ latent archetypal factors. Similarity is defined in Aitchison geometry and computed via ILR coordinates, which preserve Aitchison distances while enabling unconstrained optimization in $\mathbb{R}^{K-1}$. Under a chosen ILR basis, each coordinate corresponds to a *balance*, i.e., a log-ratio contrast between groups of archetypes, so proximity and predicted links can be interpreted in terms of relative trade-offs. Finally, subcompositional coherence ensures that restricting to a subset of components and renormalizing remains geometrically well-defined (Lemma .1), enabling principled analyses of how archetype subsets affect embeddings and predictions.

Overall, standard Euclidean graph embeddings, two nodes are close simply because the model places them nearby in an abstract vector space; the distance itself has no intrinsic meaning, and there is no principled answer to *why* the nodes are close. In our model, proximity admits a direct explanation. Nodes are close because they exhibit similar *relative mixtures* of latent factors: they emphasize the same archetypes to similar degrees relative to others. Distances in Aitchison geometry therefore quantify differences in relative trade-offs rather than arbitrary vector differences. This grounds similarity in interpretable structure and makes explanations arise directly from the geometry.

## C. Experimental Setup

**Datasets.** We evaluate on multiple undirected citation and collaboration networks: *Cora* (citation network of machine learning papers with 7 classes) (Sen et al., 2008), *Citeseer* (citation network of computer and information science papers with 6 classes) (Sen et al., 2008), *DBLP* (co-authorship network with node labels denoting research fields) (Perozzi et al., 2017), and the arXiv collaboration networks *AstroPh*, *GrQc*, and *HepTh* (Leskovec et al., 2007). We also include the social network *LastFM* (users of a music streaming service in Asia; labels correspond to 14 countries) (Rozemberczki & Sarkar, 2020). Following prior work, we treat all graphs as unweighted and undirected (citation networks are symmetrized); detailed dataset statistics are provided in the main paper.

**Baselines.** We compare AICoG against a diverse set of graph representation learning methods, including shallow embedding approaches, matrix factorization and mixed-membership models, and latent distance models. Shallow embedding methods

include NODE2VEC (Grover & Leskovec, 2016), which optimizes a skip-gram objective over biased random walks to capture network proximity, and ROLE2VEC (Ahmed et al., 2018), which uses attributed walks over structural features to learn role-based embeddings. We also include NETMF (Qiu et al., 2018), a matrix factorization method that provides an explicit factorization view of random-walk-based embeddings. Mixed-membership and factorization-based approaches include the MMSBM (Airoldi et al., 2007), a classical bilinear mixed-membership block model, and MNMF (Wang et al., 2017), which incorporates modularity regularization into nonnegative matrix factorization. We further consider simplex-based latent distance models that define node representations as mixtures over latent archetypes. These include SLIM-RAA (Nakis et al., 2023a) and HM-LDM (Nakis et al., 2022), both of which project simplex-valued representations into Euclidean space via a linear transformation to increase representational capacity. Although SLIM-RAA was originally proposed for signed networks under a Skellam likelihood, we adapt it to unsigned graphs by employing a Bernoulli likelihood. Finally, we include a SIMPLEX-EUCLIDEAN baseline, a latent distance model operating directly on the simplex using Euclidean geometry without an ILR/Aitchison transformation, in order to isolate the effect of compositional geometry.

**Tuning of the baselines.** All baseline methods were implemented using the KARATE CLUB library (Rozemberczki et al., 2020), except for NODE2VEC, which used STELLARGRAPH (Data61, 2018). We used the following hyperparameters. NETMF: iterations $= 10$; PMI power order $= 2$; negative samples $= 1$. ROLE2VEC: epochs $= 5$; window size $= 10$; walk length $= 80$; learning rate $= 0.05$. NODE2VEC: context size $= 5$; walks per node $= 10$; epochs $= 5$; $(p, q) = (1, 1)$ for link prediction and $(p, q) = (1, 1.5)$ for node classification. MNMF: clusters $= 10$; KKT penalty $= 0.2$; clustering penalty $= 0.45$; modularity regularization penalty $= 0.8$; similarity matrix parameter $= 5$. For SLIM-RAA, HM-LDM, MMSBM, and SIMPLEX-EUCLIDEAN, we optimized the negative Bernoulli log-likelihood (matching AICoG up to the model-specific log-odds parameterization) using learning rate 0.05 for 5,000 epochs, so differences are attributable only to the log-odds form.

**Link prediction.** For link prediction, we follow the widely adopted evaluation protocol of Perozzi et al. (2014); Nakis et al. (2023b). Specifically, we randomly remove $50\%$ of the edges while ensuring that the residual graph remains connected. The removed edges, together with an equal number of randomly sampled non-edges, form the positive and negative instances of the test set. The residual graph is then used to learn node embeddings. We evaluate performance on five benchmark networks, each over five runs and across multiple embedding dimensions ($D \in \{8, 16, 32, 64\}$). Table 2 reports the Area Under the Receiver Operating Characteristic Curve (AUC-ROC) (for Precision-Recall (PR-AUC) scores see supplementary). Across runs, the variance was consistently on the order of $10^{-3}$ and is omitted for readability. Following Grover & Leskovec (2016), dyadic features are constructed using binary operators (average, Hadamard, weighted-$L_1$, weighted-$L_2$) and a logistic regression classifier with $L_2$ regularization is trained to make predictions. For models that define network likelihoods predictions are obtained directly from learned rates: we use the log-odds $\eta_{ij}$ of a test pair $\{i, j\}$ to compute link probabilities, with no additional classifier required. We define three variant of our model AICoG (HB) a model trained under a Helmert basis, AICoG (LB) a model trained under a Learned basis optimized with the model, and AICoG (HB) SUBCOMP where we provide subcompoisiton coherence. Specifically, for a trained model with 65 compositions (64 dimensions) we repeatedly remove a random subset of simplex components (keeping $32, 16, 8$), reclose the remaining composition to the simplex, and recompute ILR embeddings in the reduced simplex and then evaluate link prediction reporting AUC averaged over multiple random removal masks (different seeds) for each $K'$.

# D. Additional Experiments

*Table 6.* AUC PR scores for representation sizes of 8, 16, 32, and 64 averaged over five runs.

| | AstroPh | | | | GrQc | | | | HepTh | | | | Cora | | | | DBLP | | | |
|---|---|---|---|---|---|---|---|---|---|---|---|---|---|---|---|---|---|---|---|---|
| Dimension ($D$) | 8 | 16 | 32 | 64 | 8 | 16 | 32 | 64 | 8 | 16 | 32 | 64 | 8 | 16 | 32 | 64 | 8 | 16 | 32 | 64 |
| NODE2VEC | .949 | .962 | .968 | .970 | .944 | .948 | .950 | .954 | .896 | .902 | .912 | .919 | .785 | .793 | .806 | .812 | .932 | .935 | .942 | .956 |
| ROLE2VEC | .958 | .972 | .975 | .972 | .944 | .953 | .952 | .952 | .918 | .929 | .927 | .922 | .812 | .812 | .807 | .802 | .957 | .965 | .961 | .959 |
| NETMF | .913 | .938 | .956 | .965 | .865 | .904 | .910 | .919 | .815 | .839 | .852 | .855 | .719 | .724 | .742 | .743 | .814 | .841 | .858 | .876 |
| SLIM-RAA | .970 | .975 | .974 | .974 | .953 | .956 | .959 | .961 | .920 | .929 | .934 | .935 | .827 | .815 | .816 | .821 | .964 | .968 | .970 | .971 |
| MMSBM | .894 | .914 | .920 | .920 | .850 | .850 | .845 | .853 | .801 | .784 | .780 | .786 | .693 | .697 | .683 | .685 | .812 | .810 | .814 | .810 |
| MNMF | .848 | .902 | .933 | .954 | .873 | .903 | .931 | .933 | .780 | .842 | .879 | .901 | .689 | .751 | .749 | .746 | .821 | .887 | .922 | .944 |
| HM-LDM | .952 | .954 | .956 | .960 | .953 | .955 | .956 | .958 | .900 | .899 | .906 | .909 | .841 | .843 | .839 | .844 | .910 | .900 | .913 | .939 |
| SIMPLEX-EUCLIDEAN | .873 | .868 | .869 | .881 | .857 | .847 | .841 | .839 | .765 | .753 | .755 | .769 | .759 | .748 | .743 | .736 | .702 | .689 | .704 | .757 |
| AICoG (HB) SUBCOMP | .968 | .975 | .978 | .980 | .961 | .967 | .970 | .971 | .926 | .938 | .944 | .947 | .844 | .863 | .875 | .880 | .960 | .968 | .971 | .973 |
| AICoG (HB) | .972 | .977 | .979 | .980 | .966 | .970 | .971 | .971 | .932 | .944 | .946 | .947 | .869 | .877 | .879 | .880 | .961 | .969 | .972 | .973 |
| AICoG (LB) | .972 | .977 | .979 | .980 | .964 | .970 | .971 | .971 | .932 | .944 | .946 | .946 | .869 | .877 | .880 | .881 | .961 | .969 | .972 | .973 |

**Link prediction and node classification results.**

*Table 7.* Macro-F1 scores for representation sizes of 8, 16, 32, and 64 averaged over five runs.

| Dimension ($D$) | Cora | | | | Citeseer | | | | LastFM | | | |
|---|---|---|---|---|---|---|---|---|---|---|---|---|
| | 8 | 16 | 32 | 64 | 8 | 16 | 32 | 64 | 8 | 16 | 32 | 64 |
| NODE2VEC | .755 | .761 | .785 | .801 | .503 | .565 | .593 | .632 | .717 | .779 | .786 | .794 |
| ROLE2VEC | .764 | .781 | .786 | .791 | .572 | .606 | .604 | .623 | .720 | .771 | .775 | .775 |
| NETMF | .721 | .744 | .766 | .782 | .502 | .563 | .593 | .619 | .431 | .573 | .712 | .754 |
| SLIM-RAA | .561 | .601 | .669 | .711 | .432 | .478 | .477 | .526 | .385 | .572 | .627 | .697 |
| MMSBM | .125 | .279 | .459 | .567 | .259 | .354 | .445 | .468 | .091 | .244 | .315 | .522 |
| MNMF | .460 | .591 | .662 | .703 | .337 | .400 | .485 | .545 | .298 | .467 | .626 | .714 |
| HM-LDM | .676 | .700 | .762 | .797 | .513 | .511 | .566 | .611 | .604 | .668 | .705 | .740 |
| SIMPLEX-EUCLIDEAN | .369 | .370 | .445 | .386 | .338 | .304 | .325 | .360 | .165 | .211 | .282 | .360 |
| AICoG (HB) SUBCOMP | .648 | .766 | .803 | .815 | .473 | .571 | .631 | .667 | .676 | .779 | .675 | .794 |
| AICoG (HB) | .759 | .794 | .815 | .815 | .565 | .596 | .643 | .667 | .755 | .789 | .793 | .794 |
| AICoG (LB) | .767 | .792 | .812 | .816 | .572 | .600 | .633 | .667 | .753 | .791 | .793 | .795 |

**Subcompositional robustness under random removals.** We evaluate whether the learned representations remain predictive under subcomposition. Starting from learned simplex embeddings $\mathbf{x}_i \in \Delta^{K-1}$ (obtained by applying a softmax to the learned logits), we sample a random subset $S \subseteq \{1, \ldots, K\}$ of size $|S| = K'$ (using multiple random seeds). For each sampled $S$, we form the subcomposition by dropping components outside $S$ and reclosing, $\mathbf{x}_i^{(S)} = \mathbf{x}_{i,S} / \sum_{\ell \in S} x_{i\ell}$, and compute ILR coordinates in the reduced simplex, $\mathbf{y}_i^{(S)} = \mathrm{ILR}(\mathbf{x}_i^{(S)}) \in \mathbb{R}^{K'-1}$ (Helmert basis). We score candidate edges using the same latent-distance decoder, $s_{ij}^{(S)} = -\alpha_S \|\mathbf{y}_i^{(S)} - \mathbf{y}_j^{(S)}\|_2 + \gamma_i + \gamma_j$, and report AUC and AUPRC on a fixed evaluation set of positive and negative edges. To keep the distance and bias terms comparable across different $K'$, we optionally rescale distances via $\alpha_S$ using median-distance calibration on the evaluation pairs. Results are averaged over many random subsets $S$ for each $K'$.

