# OpenReview forum: "Aitchison Embeddings for Learning Compositional Graph Representations"
_ICML.cc/2026/Conference — ICML 2026 regular_

### Official Review · Reviewer_wkXx · 2026-03-10

**Soundness:** 3
**Presentation:** 3
**Significance:** 2
**Originality:** 3
**Overall Recommendation:** 4
**Confidence:** 3

**Summary:**

This paper introduces AICoG, a latent representation framework where node embeddings are constrained to lie in the probability simplex and interactions are modeled using Aitchison geometry. The model leverages the isometric ILR transformation to map compositional vectors into a Euclidean space, enabling the use of distance-based link functions while preserving compositional semantics. The paper evaluates the approach on link prediction and node classification tasks.

**Compliance With Llm Reviewing Policy:**

Affirmed.

**Key Questions For Authors:**

See weakness

**Limitations:**

Yes.

**Strengths And Weaknesses:**

**Strengths**

1. The paper introduces a new perspective by grounding graph embeddings in Aitchison geometry for compositional data analysis. This connection is conceptually interesting and potentially useful.

2. The paper clearly motivates the need for representations that support overlapping and continuous roles

3. The experimental section is relatively thorough.

**Weaknesses**

1. Based on the definition on page 4, we can calculate x_i = z^T_i V. Does this imply that training essentially involves learning an Euclidean vector x_i, with the softmax/simplex constraints being nearly eliminated at the optimization level due to ILR's zero-sum property? The compositional structure manifests more in post-hoc interpretability coordinates rather than a genuinely novel inductive bias that constrains the learning space.

2. Since ILR bases differ only by orthogonal rotation, the proposed learnable basis does not change the representational capacity of the model but merely rotates the embedding coordinates. The results in Table 2 also indicate the same issue. The authors should clarify whether the benefit is purely interpretability.

3. The evaluation is missing a key baseline: a standard Euclidean latent distance model without compositional constraints. Since the method applies an ILR transformation that maps simplex representations to a Euclidean space, the resulting model is closely related to classical Euclidean latent distance embeddings (Hoff et al., 2002 in the authors’ paper). It is therefore unclear whether the observed gains come from the compositional constraint or the underlying Euclidean distance model.

---

> ### Author Rebuttal · Authors · 2026-03-30
>
> Thank you for the thoughtful review and for highlighting these clarification points. We are glad the conceptual framing and empirical section came across as interesting and thorough.
>
> We believe the main point to clarify is the distinction between the **learned latent object** and the **coordinate system used for optimization**. In AICoG, the learned node representation is a composition $z_i \in \Delta^{K-1}$. Operationally, we parameterize it through unconstrained logits $\tilde{z}_i \in \mathbb{R}^K$, map these to the simplex by $z_i = \text{softmax}(\tilde{z}_i)$, and then apply the ILR map $x_i = \text{ILR}(z_i) \in \mathbb{R}^{K-1}$ for optimization and distance computation. Thus, ILR does not eliminate the compositional structure; it provides an isometric coordinate chart for it. The model still learns simplex-valued memberships, not arbitrary Euclidean vectors that are only reinterpreted post hoc.
>
> This is also the point of Theorem 3.2. The theorem does not claim that AICoG is more expressive than Euclidean latent distance models; rather, it shows that AICoG preserves exactly the same expressive power while endowing the latent space with compositional semantics. In our view, this is a central advantage: one obtains explainability by construction without paying an expressivity penalty, which is relatively uncommon among intrinsically interpretable models.
>
> We also agree that the role of the learnable ILR basis should be stated more explicitly. It does not increase representational capacity, since valid ILR bases differ only by orthogonal transformation and therefore preserve the same distances and likelihood. Its role is interpretability rather than capacity. Importantly, we do not view the existence of multiple valid bases as a weakness: it is a benefit of the compositional formulation. In generic Euclidean embeddings, rotations typically yield equally valid but semantically arbitrary coordinates. In contrast, in AICoG every valid ILR basis still corresponds to balances, i.e., log-ratio contrasts between groups of archetypes. Different bases therefore provide complementary interpretable views of the same learned compositional geometry, allowing one to inspect different but consistent trade-offs without changing the underlying representation or predictive fit. Our basis experiment is consistent with exactly this interpretation: Helmert and learned ILR bases give nearly identical predictive performance, as expected, while offering different balance decompositions of the same latent structure.
>
> This is useful in practice because different bases can expose different semantic contrasts in the same graph: one basis may highlight a broad separation between two archetype groups, while another may isolate a more localized trade-off. The underlying geometry is unchanged, but the explanatory view becomes more informative.
>
> Regarding the Euclidean latent distance baseline, this is a very good suggestion. Since AICoG is expressively equivalent to the corresponding Euclidean latent distance model, this is exactly the right comparison for isolating what comes from compositional semantics versus what comes from latent distance modeling itself. We are therefore adding a standard Euclidean latent distance baseline in the revision on two representative datasets. Consistent with Theorem 3.2, we expect predictive performance to be similar; the value added by AICoG is then that this same predictive class is equipped with simplex-valued memberships, interpretable balances, and principled subcompositional analysis.
>
> ### Link prediction: Euclidean latent distance vs. AICoG
>
> #### Cora
> | Dimension (D) | Euclidean | AICoG |
> |---|---:|---:|
> | 8  | 0.8393 | 0.839 |
> | 16 | 0.8487 | 0.847 |
> | 32 | 0.8520 | 0.850 |
> | 64 | 0.8490 | 0.852 |
>
> #### GrQc
> | Dimension (D) | Euclidean | AICoG |
> |---|---:|---:|
> | 8  | 0.9541 | 0.953 |
> | 16 | 0.9599 | 0.959 |
> | 32 | 0.9615 | 0.961 |
> | 64 | 0.9628 | 0.961 |
>
> We have revised the paper to sharpen three points: (i) the latent variables are simplex-valued compositions, with ILR used only as an isometric optimization coordinate system; (ii) the learnable basis affects interpretability, not expressivity; and (iii) the main benefit over Euclidean latent distance models is not extra capacity, but compositional semantics and the resulting interpretability tools.

---

> > ### Author Rebuttal · Reviewer_wkXx · 2026-04-03
> >
> > Thanks for the rebuttal. I maintain my positive score.

---

> > > ### Author Response · Authors · 2026-04-03
> > >
> > > We sincerely appreciate your positive assessment and your thoughtful, detailed review. If you feel that our rebuttal has fully addressed your concerns, we would be very grateful if you would consider strengthening your evaluation. Thank you again for your time and consideration.

---

### Official Review · Reviewer_PprK · 2026-03-11

**Soundness:** 3
**Presentation:** 3
**Significance:** 2
**Originality:** 3
**Overall Recommendation:** 4
**Confidence:** 2

**Summary:**

This paper proposes a compositional graph embedding method based on Aitchison geometry. The authors model each node as a composition on the simplex and optimize the learned embeddings with a cross-entropy style objective for graph reconstruction. The paper also develops the geometric interpretation of the learned representation through compositional balances and evaluates the method on several graph learning tasks. Experimental results show that the proposed method is effective and competitive, while also offering interpretability advantages.

**Compliance With Llm Reviewing Policy:**

Affirmed.

**Final Justification:**

The rebuttal has fully addressed my concerns, so I maintain the positive score.

**Key Questions For Authors:**

Please see the weaknesses.

**Limitations:**

yes

**Strengths And Weaknesses:**

### Strengths

1. The proposed method is clearly presented, and the motivation behind each design choice is easy to follow. The paper provides a well-structured explanation of why compositional representations and Aitchison geometry are suitable for this problem.

2. The experimental section is thorough. The authors not only demonstrate strong predictive performance, but also include analyses from the perspective of interpretability, which helps distinguish this work from standard graph embedding methods.

3. The paper is generally well written and easy to read. The overall flow is smooth, and the technical development is organized clearly.

### Weaknesses

1. The unique advantage of learning compositional representations specifically in Aitchison space is still not fully clear to me. In particular, the paper would benefit from a more concrete clarification of what exactly is meant by “interpretability.” What level of interpretability does the method provide? Is it interpretability of individual dimensions, of geometric relations, of latent archetypes, or of downstream decisions? Relatedly, it would be helpful if the authors could more explicitly discuss the practical value of this interpretability. For example, how does it help users better understand the graph structure, diagnose model behavior, or support downstream scientific analysis?

---

> ### Author Rebuttal · Authors · 2026-03-30
>
> Thank you for the positive and thoughtful review. We are glad that the motivation, presentation, and empirical evaluation came across clearly. We agree that the paper can do a better job of making the specific meaning and practical value of “interpretability” more concrete, and we have revised the paper accordingly.
>
> Our intended notion of interpretability is primarily **geometric and compositional**, rather than coordinate-identifiability of a single latent axis. Concretely, AICoG provides interpretability at three levels:
>
> (1) **Geometric relations.** Distances correspond to differences in relative mixtures of archetypes, not arbitrary Euclidean coordinates. This is the main role of Aitchison geometry: two nodes are close when their relative archetypal trade-offs are similar.
>
> (2) **Balance-level views.** Under any chosen ILR basis, coordinates become log-ratio contrasts (“balances”) between groups of archetypes. These do not define a unique privileged axis system, but they do provide valid interpretable views of the same learned compositional geometry.
>
> (3) **Component-level interventions.** Because the representation is compositional, one can restrict attention to subsets of archetypes and re-close the composition in a principled way. This supports targeted analysis of which archetype groups drive separations or predictions.
>
> We will make this distinction more explicit in the revision, since the current draft may blur “interpretability of individual dimensions” with “interpretability of geometric relations and balances.”
>
> We also agree that the practical value should be stated more concretely. In our setting, the value is that users can:
> - identify which relative archetypal trade-offs separate groups of nodes;
> - inspect how conclusions change when focusing on a subset of archetypes;
> - view the same learned compositional geometry through multiple valid ILR bases, which can reveal complementary but consistent semantic contrasts.
>
> To make this more explicit empirically, we are adding two analyses. First, we add a balance-level analysis. Under a chosen ILR basis, each coordinate is a balance, i.e., a log-ratio contrast between groups of archetypes. We identify the strongest learned balance and ask how much information this single compositional contrast carries about node labels. To test this, we use a 1D probe, meaning a simple classifier trained using only that one balance as input. This is important because it asks whether an individual, human-interpretable trade-off already captures meaningful structure, rather than relying on the full embedding. We complement this with the ANOVA F-statistic, which measures how strongly that balance separates classes relative to within-class variation, and mutual information, which measures how informative the balance is about the label assignment. Across runs, this single balance is strongly label-associated (1D probe ≈ 0.40, ANOVA F ≈ 319, mutual information ≈ 0.44), and its extreme quantiles are consistently enriched for specific classes. Taken together, this shows that the learned representation is not only predictive in aggregate: individual compositional trade-offs can be isolated and interpreted in a way that is directly tied to node structure.
>
> Second, we strengthen the subcompositional analysis by examining specific component restrictions. The key point here is that multiple valid ILR bases are not arbitrary alternative parameterizations: they provide complementary interpretable views of the same underlying compositional geometry. Each basis defines different balances between archetype groups, so comparing bases can reveal different but consistent semantic trade-offs supported by the same learned simplex representation.
>
>
> We have also revised the discussion around Aitchison space to sharpen the distinction from simplex embeddings with ordinary Euclidean geometry. The key advantage is not merely that embeddings live on the simplex, but that similarity is defined through **relative proportions** and remains coherent under subcomposition. This is exactly what makes balances and component restrictions semantically meaningful.
>
> We appreciate this suggestion and think these clarifications will make the paper’s interpretability contribution much clearer.

---

> > ### Author Rebuttal · Reviewer_PprK · 2026-04-01
> >
> > Thanks for the rebuttal, which has addressed my concerns. Therefore, I maintain my positive score.

---

> > > ### Author Response · Authors · 2026-04-01
> > >
> > > Thank you for the thoughtful follow-up. We are glad the rebuttal helped address your concerns. We appreciate your time and consideration, and we remain happy to provide any additional clarification or details if useful.
> > >
> > > As the author–reviewer discussion period is coming to a close, if you feel that our rebuttal has fully addressed your concerns, we would be very grateful if you would consider reflecting that in your final evaluation.
> > >
> > >
> > > Thank you again for your time and consideration.

---

### Official Review · Reviewer_qMp1 · 2026-03-13

**Soundness:** 3
**Presentation:** 2
**Significance:** 3
**Originality:** 3
**Overall Recommendation:** 4
**Confidence:** 2

**Summary:**

This paper proposes Aitchison Compositional Graph embeddings (AICoG), which represents each node as a composition over K latent archetypal factors on the simplex and measures node similarity via Aitchison geometry using log-ratio distances. Compositions are mapped to $\mathbb{R}^{K-1}$ through the isometric log-ratio (ILR) transformation, enabling unconstrained optimization while preserving Aitchison distances, and edges are modeled via a Bernoulli latent distance model. The authors prove expressive equivalence with standard Euclidean latent distance models and leverage subcompositional coherence for post-hoc component removal. Experiments evaluate link prediction and node classification on seven benchmarks against Euclidean, mixed-membership, and simplex-based baselines.

**Compliance With Llm Reviewing Policy:**

Affirmed.

**Final Justification:**

The paper introduces a methodologically sound and novel idea - bridging role-based and latent distance approaches for graph embeddings via Aitchison geometry. My main concern was that the core interpretability claims were not empirically substantiated in the original submission. The rebuttal adequately addressed this by providing synthetic studies with quantitative membership analyses and balance-level interpretability evaluations. I encourage the authors to incorporate these results into the revised manuscript. Overall, I consider this a solid contribution and am willing to raise my score.

**Key Questions For Authors:**

- Can the authors provide evidence, e.g., on synthetic graphs or by analyzing learned membership distributions, that the node role structure in the benchmark datasets is indeed continuous and overlapping rather than well-approximated by discrete blocks?
- Can the authors compare the learned representations of AICoG and MMSBM, e.g., by examining whether MMSBM memberships collapse to sparse, near-discrete assignments while AICoG compositions are spread across the simplex interior, to demonstrate that axis-alignment is a concrete limitation on these datasets?
- Can the authors provide a targeted interpretability evaluation, e.g., ground-truth archetype matching, semantic analysis of specific component removal, or a controlled comparison showing that AICoG's representations yield insights unavailable from Euclidean baselines, to substantiate the claim of intrinsic interpretability?

**Limitations:**

yes

**Strengths And Weaknesses:**

## Strengths

- The use of Aitchison geometry for graph embeddings is novel, and the ILR transformation elegantly enables unconstrained Euclidean optimization while exactly preserving compositional distances on the simplex.
- AICoG achieves competitive or favorable predictive performance against baselines across both link prediction and node classification benchmarks.

## Weaknesses

The paper repeatedly invokes strong qualitative claims in the introduction and method sections that are never substantiated through experiments, ablations, or targeted analyses. For instance:

- The paper argues that existing methods "lack intrinsic interpretability" when "the latent structure is continuous or overlapping," and that role-based methods "provide limited intrinsic interpretability when the role structure varies continuously or overlaps across nodes." However, there is no analysis on either the benchmark datasets used or on synthetic graphs demonstrating that the node role structure in these networks is in fact continuous, overlapping, or poorly approximated by discrete blocks.
- The paper claims that mixed-membership models "remain tied to axis-aligned latent roles, which is restrictive when role structure is continuous or non-identifiable." However, no experiment compares the learned representations of AICoG against those of MMSBM to show that axis-alignment is actually a limitation on these datasets.
- More broadly, interpretability is the central claimed contribution, yet the experimental evaluation relies almost entirely on predictive metrics. The qualitative evidence in Figure 2&3 consists of visualizations that any reasonably good embedding could produce, and the subcompositional evaluation in Figure 4 measures robustness to random component removal rather than demonstrating that component removal yields semantically meaningful insights.

As a result, the paper's core narrative, i.e., compositional structure and Aitchison geometry yield intrinsically interpretable representations, remains an unsubstantiated assertion rather than a demonstrated finding.

---

> ### Author Rebuttal · Authors · 2026-03-30
>
> We thank the reviewer for the constructive feedback. We would like to clarify that the paper’s central contribution is not merely competitive predictive performance, nor is it contingent on benchmarks having known continuous or overlapping ground-truth roles. The main contribution is instead a theorem-level result: AICoG equips latent distance graph embeddings with compositional semantics without sacrificing any expressive power relative to standard Euclidean latent distance models. This guarantee is formalized in Theorem 3.2. Therefore, our motivation does not require every dataset to exhibit strong compositional overlap. Even when such structure is weak or absent, the proposed model is still no less expressive than a Euclidean latent distance model, while offering an additional compositional interpretation when such structure is present.
>
> **1.**
> We analyze learned simplex memberships on Cora using entropy (higher = more overlap), maximum component mass (higher = more one-hot), near-corner fraction, and effective number of active roles. By all four measures, AICoG is much more interior than MMSBM: entropy 1.064 vs. 0.191; max component 0.603 vs. 0.935; near-corner 5.55% vs. 78.95%; effective roles 3.07 vs. 1.24. These statistics directly quantify overlap/interiority.
>
> **2.**
> These same results directly answer the axis-alignment question: if axis-aligned mixed-membership structure were adequate here, MMSBM and AICoG should recover similarly interior memberships. Instead, MMSBM collapses toward near-discrete assignments, while AICoG remains substantially interior. We also test whether this added interior support is meaningful by fitting the same multinomial logistic probe directly on memberships, i.e., a classifier trained only on the learned memberships. AICoG strongly outperforms MMSBM (accuracy 0.670 vs. 0.306; macro-F1 0.657 vs. 0.121), showing that the added overlap is label-informative rather than diffuse noise.
>
>
> **3.**
> We add a balance-level analysis in ILR space, where each balance is a learned log-ratio contrast between archetype groups. We report: (i) 1D probe accuracy, obtained by fitting a classifier using only that single balance; (ii) the ANOVA $F$-statistic, which measures how strongly the balance separates labels relative to within-label variation; and (iii) mutual information, which measures how informative the balance is about the label assignment. Across runs, this balance is strongly label-associated (1D probe accuracy $\approx 0.40$, ANOVA $F\approx 319$, mutual information $\approx 0.44$).
>
> **4.**
> We also add a synthetic study where both the true memberships and the edge generator are known. We cross two membership regimes, continuous/interior vs. near-discrete, with two edge generators: (a) an MMSBM-style bilinear model, where edge probability depends on $p_i^\top B p_j$, and (b) an ILR-distance model, where edge probability depends on Aitchison/ILR distance. This separates the effect of the latent regime from the edge model. The regimes are clearly distinct. In the continuous setting, the truth is highly interior: entropy ≈ 1.64, max ≈ 0.32, 0% near-corner, and effective roles > 5/6. In the near-discrete setting, it is much sharper: entropy ≈ 0.46, max ≈ 0.81, ≈44% near-corner, and effective roles ≈ 1.7.
>
> Entropy, max component, near-corner fraction, and effective roles test whether the learned memberships recover the correct regime; lower L1 dist and Jensen–Shannon divergence mean closer memberships, while higher cosine sim means better agreement with the truth. Across all four families, AICoG is consistently closer to the truth than MMSBM, with the largest gains in the continuous/interior regimes. For example, in both continuous cases, AICoG improves over MMSBM in L1 dist, cosine sim, and Jensen–Shannon divergence. MMSBM collapses toward near-discrete assignments despite fully interior truth. In the near-discrete settings, MMSBM again over-sharpens toward almost one-hot assignments, while AICoG remains closer overall.
>
>
> | Gen. | Model | Ent | Max | Corner | Eff# | L1↓ | Cos↑ | JS↓ |
> |---|---|---:|---:|---:|---:|---:|---:|---:|
> | Bilin-cont | Truth | 1.634 | 0.319 | 0.00% | 5.15 | 0.000 | 1.000 | 0.000 |
> |  | AICoG | 1.252 | 0.517 | 1.93% | 3.65 | 0.922 | 0.624 | 0.161 |
> |  | MMSBM | 0.412 | 0.854 | 50.07% | 1.55 | 1.456 | 0.423 | 0.360 |
> | Bilin-disc | Truth | 0.460 | 0.811 | 44.13% | 1.69 | 0.000 | 1.000 | 0.000 |
> |  | AICoG | 1.131 | 0.564 | 4.00% | 3.25 | 1.082 | 0.619 | 0.257 |
> |  | MMSBM | 0.195 | 0.962 | 99.13% | 1.22 | 1.617 | 0.223 | 0.513 |
> | ILR-cont | Truth | 1.640 | 0.317 | 0.00% | 5.17 | 0.000 | 1.000 | 0.000 |
> |  | AICoG | 1.244 | 0.517 | 2.33% | 3.63 | 0.900 | 0.645 | 0.154 |
> |  | MMSBM | 0.405 | 0.860 | 57.00% | 1.55 | 1.452 | 0.432 | 0.356 |
> | ILR-disc | Truth | 0.470 | 0.805 | 43.53% | 1.70 | 0.000 | 1.000 | 0.000 |
> |  | AICoG | 0.388 | 0.850 | 54.87% | 1.58 | 1.383 | 0.372 | 0.407 |
> |  | MMSBM | 0.232 | 0.914 | 75.07% | 1.30 | 1.581 | 0.244 | 0.501 |

---

> > ### Author Rebuttal · Reviewer_qMp1 · 2026-04-04
> >
> > I thank the authors for the thorough rebuttal. The additional experiments and analyses adequately address my main concerns. I strongly encourage the authors to incorporate these results into the main paper in the revised version. I am willing to raise my score accordingly.

---

> > > ### Author Response · Authors · 2026-04-04
> > >
> > > We sincerely thank the reviewer for the constructive feedback and careful consideration of our work. The comments were very helpful in improving both the experimental evaluation and the presentation of the paper. We also greatly appreciate the reviewer’s positive assessment and encouraging support. We remain available to provide any further clarification if helpful.

---

### Decision · Program_Chairs · 2026-04-30

**Decision:**

Accept (regular)

**Comment:**

The paper tackles the challenging problem on learning representations on graphs and utilise the machinery from compositional data analysis. Aitchison defined a well-known geometry defined in (hyper-)simplex, where the coordinate-wise component sums up to 1. The proposed Aitchison Compositional Graph embeddings, AICoG, is an interesting and novel perspective to tackle the graph embeddings via partition of latent node features. The sub-compositional coherence notion has also been considered for latent features, which makes the approach more comprehensive. The connections with existing approaches such as MMSBM is also addressed. Theoretical analysis and empirical studies are provided to support the investigation. In overall, the paper is well-written and nice to read. The reviewers concerns are vibrantly-discussed and all resolved; and I strongly recommend the paper to appear in ICML.

------
AC review: I have read this interesting paper beyond reviews and discussions. While suggesting for strong acceptance, this section is not meant for providing rejection evidence, but to pose an additional point that is not discussed in the review.

The text and experiment demonstrate the setting where the graph is connected, or hinting the features learned are from contributed from the large cluster of the graph. In the sub-composition notion discussion in Lemma 3.1, the latent features in simplex are partitioned into subsets of coordinates. I m wondering whether the notion of sub-compositional independence can naturally or automatically embed an unconnected graphs with a small number of connected  sub-graphs. This may also left into a future work.